# Mathematical Contact Tracing Models for the COVID-19 Pandemic: A Systematic Review of the Literature

**DOI:** 10.3390/healthcare13080935

**Published:** 2025-04-18

**Authors:** Honoria Ocagli, Gloria Brigiari, Erica Marcolin, Michele Mongillo, Michele Tonon, Filippo Da Re, Davide Gentili, Federica Michieletto, Francesca Russo, Dario Gregori

**Affiliations:** 1Unit of Biostatistics, Epidemiology and Public Health, Department of Cardiac, Thoracic, Vascular Sciences, and Public Health, University of Padova, Via Loredan 18, 35122 Padova, Italy; honoria.ocagli@unipd.it (H.O.);; 2Directorate of Prevention, Food Safety, Veterinary Public Health, Veneto Region, 30123 Venice, Italy; gloria.brigiari@studenti.unipd.it (G.B.); michele.mongillo@regione.veneto.it (M.M.); michele.tonon@regione.veneto.it (M.T.); filippo.dare@regione.veneto.it (F.D.R.); davide.gentile@regione.veneto.it (D.G.); federica.michieletto@regione.veneto.it (F.M.); francesca.russo@regione.veneto.it (F.R.)

**Keywords:** contact tracing, COVID-19, mathematical models, systematic review

## Abstract

**Background**: Contact tracing (CT) is a primary means of controlling infectious diseases, such as coronavirus disease 2019 (COVID-19), especially in the early months of the pandemic. **Objectives**: This work is a systematic review of mathematical models used during the COVID-19 pandemic that explicitly parameterise CT as a potential mitigator of the effects of the pandemic. **Methods**: This review is registered in PROSPERO. A comprehensive literature search was conducted using the PubMed, EMBASE, Cochrane Library, CINAHL, and Scopus databases. Two reviewers independently selected the title/abstract, full text, data extraction, and risk of bias. Disagreements were resolved through discussion. The characteristics of the studies and mathematical models were collected from each study. **Results**: A total of 53 articles out of 2101 were included. The modelling of the COVID-19 pandemic was the main objective of 23 studies, while the remaining articles evaluated the forecast transmission of COVID-19. Most studies used compartmental models to simulate COVID-19 transmission (26, 49.1%), while others used agent-based (16, 34%), branching processes (5, 9.4%), or other mathematical models (6). Most studies applying compartmental models consider CT in a separate compartment. Quarantine and basic reproduction numbers were also considered in the models. The quality assessment scores ranged from 13 to 26 of 28. **Conclusions**: Despite the significant heterogeneity in the models and the assumptions on the relevant model parameters, this systematic review provides a comprehensive overview of the models proposed to evaluate the COVID-19 pandemic, including non-pharmaceutical public health interventions such as CT. Prospero Registration: CRD42022359060.

## 1. Introduction

Coronavirus disease 2019 (COVID-19), caused by the highly contagious Severe Acute Respiratory Syndrome Coronavirus 2 (SARS-CoV-2), is a viral illness [1]. On 23 December 2020, the novel coronavirus, first identified in Wuhan, China, caused 6,656,601 deaths worldwide and 651 million confirmed cases [2]. Emerging infectious diseases, such as COVID-19, require rapid response and targeted control measures to prevent massive spread at the beginning of the pandemic. Non-pharmaceutical public health interventions (NPHIs) were essential, especially in the first months of the pandemic, as there was no reliable vaccine or treatment to control or treat the disease [3]. These measures were designed to prevent infection transmission, including maintaining physical distance, wearing masks, quarantine, isolation, and contact tracing (CT). A recent systematic review and meta-analysis on the effectiveness of NPHIs reported a decrease in daily case growth rates, daily death growth rates, and COVID-19 reproduction numbers of −4.68% (95% CI, −6.94, −2.78), −4.8% (95% CI, −8.34, −1.40), −1.90% (95% CI, −2.23, −1.58), and −16.5% (95% CI, −19.68, −13.32) [3]. CT is defined as the identification, evaluation, and management of people exposed to a disease to prevent subsequent transmission [4]. CT is the primary means of controlling infectious diseases, such as tuberculosis, AIDS, and other sexually transmitted diseases [5,6]. However, it also played a crucial role during some epidemics, such as the Ebola outbreak in Africa in 2014 [7]. CT focuses on potential next-generation cases by identifying individuals who have had contact with infectious individuals. Individuals who have been exposed but are not yet infectious should be identified and isolated to prevent the spread of the disease [6]. CT acts at the individual and population levels; therefore, infected persons are diagnosed earlier. Thus, it becomes possible to identify and stop transmission chains, leading to a decrease in the effective number of reproductions [8]. Therefore, the main effect of these interventions was a decrease in the effective transmissibility of infection [9]. Consequently, this implies a strong relationship between CT and the transmission patterns. CT must be integrated with other NPHI measures, such as quarantine and screening, as reported by Girum et al. [10] to be an effective measure. CT can be implemented using both forward and backward approaches. Forward contact tracing focuses on identifying and isolating individuals who may have been infected by the index case during their infectious period, typically starting two days before symptom onset or diagnosis. Backward tracing focuses on identifying the source of infection and other cases linked to the same exposure event. This approach may also uncover additional secondary cases, especially when the index case is infectious several days before symptom onset [4,11].

Several mathematical models have been implemented to study the disease transmission process during the COVID-19 pandemic. A recent review by Saleem et al. [12] found that machine learning techniques, deep learning, and compartmental models helped to understand the epidemiology and forecast the impact of the epidemic. Deep learning and machine learning algorithms are mainly used to forecast, classify, and automatically detect. Compartmental models have helped to understand the epidemiological characteristics of the pandemic [12]. Mathematical models have been designed to describe the transmission and propagation of the COVID-19 pandemic [13]. They can help target control measures, such as CT, with increased efficiency and understand key transmission parameters and NPHI [13]. Researchers have also considered a hybrid model (that combines the demand for health resources and disease transmission). These models not only predict health service utilisation, deaths, confirmed cases, and flattening of the curve, but also consider the impact of CT and other non-pharmaceutical interventions (NPHI) on the dynamics of the epidemic. System dynamics models, as highlighted in the work of Fair et al. [14] and Yusoff [15], provide an interconnected approach to optimise pandemic response strategies. These models, coupled with computer experimental design and statistical analysis, produce rapid and quantitative results for decision support, demonstrating that widespread testing, contact tracing, and quarantine play an essential role in curtailing the pandemic by identifying asymptomatic individuals in the population [14,15]. However, only some of these models explicitly modelled the impact of CT and other NPHIs on epidemics. Considering NPHIs in models can help evaluate interventions based on the scenarios considered.

This work aims to systematically review mathematical models used in the COVID-19 pandemic that considered CT in their parameters as important in mitigating the effects of COVID-19. By focusing on CT, we aim to shed light on its distinctive contribution to epidemic control, recognising its complementary role alongside other interventions in multifaceted pandemic responses.

## 2. Materials and Methods

This systematic review was based on the Preferred Reporting Items for Systematic Reviews and Meta-Analyses (PRISMA) statement [16]. This review is registered in PROSPERO (registration number CRD42022359060).

### 2.1. Eligibility Criteria

This research included peer-reviewed articles that used a mathematical model to describe COVID-19, incorporating CT as one of its parameters. Articles that used existing mathematical models with parameter modifications were included. Studies that targeted a specific population rather than the community and non-English language studies were excluded. Reviews, comments, letters, editorials, meta-analyses, and non-peer-reviewed articles were also excluded.

### 2.2. Search Strategy and Information Source

Comprehensive literature research was conducted in the Medline (via PubMed), EMBASE (through Ovid), Cochrane Library (through Ovid), CINAHL, and Scopus databases. The last search was updated on 15 May 2022. This date approximately coincides with the end of the acute phase of the COVID-19 pandemic in many countries, when large-scale interventions, such as contact tracing, began to be phased out or significantly modified. The reference lists of other systematic reviews and included studies were screened to retrieve relevant articles. The search string was based on three main concepts: COVID-19, CT, and statistical models, combined using the Boolean operator AND. Keywords within each concept were associated using the Boolean operator OR. Specifically, we included the following keywords: “contact tracing” for the first concept; “modeling”, “models”, “model”, “statistical models”, and “mathematical model” for the second; and the third concept, terms such as “COVID-19”, “COVID-19 vaccines”, “COVID-19 serotherapy”, “COVID-19 nucleic acid testing”, “COVID-19 serological testing”, “COVID-19 testing”, “SARS-CoV-2”, “Severe Acute Respiratory Syndrome Coronavirus 2”, “NCOV”, “2019 NCOV”, “COVID-19 breakthrough infections”, “spike protein SARS-CoV-2”, “COVID-19 vaccine booster shot”, “SARS-CoV-2 variants”, “coronavirus”, and “COV”. The search string was first created for PubMed and then adapted to other databases. The complete PubMed search strategy is available in Appendix B, Table A1.

### 2.3. Study Selection

Two independent reviewers performed title/abstract and full-text screening. Disagreements were resolved through discussion and consultation with another reviewer when consensus was not reached. All screenings were performed on the Covidence platform, and duplicates were automatically removed [17]. The flowchart (Figure 1) shows the study selection process [18].

### 2.4. Data Extraction

Data extraction was performed using Covidence by two reviewers. The two reviewers prepared a data extraction table, tested it on three articles, and finalised it according to the information retrieved from the pilot articles. The following information was collected: general information (title, first author, year of publication, country, aim of the study, study design, setting, and number of participants), characteristics of the proposed model (parameters in use, key relevant assumption, basic reproduction number, incubation period, latent period, infectious period, serial interval), contact definition, and epidemiological and intervention parameters of the comparator incorporated in the model. Furthermore, we recorded whether the studies implemented forward or backward contact tracing strategies. Forward contact tracing refers to the identification of individuals potentially infected by a known case, while backward contact tracing aims to identify the source of infection by tracing upstream [4,11].

### 2.5. Synthesis Methods

A descriptive overview of the retrieved articles will be provided, focusing on the study characteristics and parameters included in the models.

### 2.6. Risk of Bias

The quality of the studies was evaluated using the modified tool proposed in a systematic review by Harris et al. [19]. The instrument has 14 criteria and is used in epidemiological modelling studies. The study could be considered of low (score < 14), medium (14–18), high (19–22), or very high (>22) quality.

## 3. Results

A total of 2101 articles were identified (Figure 1), and after removing 1078 duplicates, 53 articles were included in this review. In the full-text screening, the most common reasons for exclusion were wrong outcome (n = 178), meaning the study did not evaluate the impact of contact tracing or did not include it as a parameter in the model; wrong study design (n = 26); and no new model (n = 23) (Figure 1).

### 3.1. Study Characteristics

The outcomes of each study are summarised in Table A2. The modelling of the COVID-19 pandemic was the main objective of 23 studies, forecasting the transmission of COVID-19 in different scenarios, including contact tracing and assessment of CT effectiveness [20,21,22,23,24,25,26,27,28,29,30,31,32,33,34,35,36,37,38,39,40,41,42]. Twenty-four studies focused on assessing the effects of CT or other control measures [20,21,22,23,24,25,26,27,28,30,31,32,33,34,35,36,37,38,39,41,42,43,44,45]. The remaining articles evaluate the forecast transmission of COVID-19 [46,47,48,49,50,51].

The study characteristics of the included studies are reported in Appendix A. The number of populations considered varied from 101 in the study by James in New Zealand [52] to more than 83 million individuals in the study by Grimm et al., which modelled the case study on the German population [32]. The extracted studies considered both real data (26, 49%) [20,22,23,24,26,27,28,32,37,41,42,46,47,48,49,51,52,53,54,55,56,57,58,59,60,61] and simulated data (27, 51%) [21,25,29,30,31,33,34,35,36,38,39,40,43,44,45,50,62,63,64,65,66,67,68,69,70,71,72]. The countries are represented as follows: United States (10, 18.9%) [23,27,30,31,35,36,41,57,58,63], United Kingdom (5, 9.4%) [33,37,53,56,62], China (4, 7.5%) [24,42,46,64], Spain (3, 5.66%) [28,34,54], South Corea (3, 5.66%) [25,40,43], Italy (2, 3.8%) [48,55], Brazil [20], Germany [32], India [65], Indonesia [51], Malaysia [22,47], New Zeland [52], mixed [49], Switzerland [21].

### 3.2. Ccontact Tracing

The studies included in the review showed a high degree of heterogeneity in the implementation of contact tracing, both manual (mCT) and digital (16, 30.3%) (Appendix A). Nineteen articles [25,27,28,33,35,36,37,45,46,51,55,57,61,62,63,64,66,69,71] considered different scenarios, mainly different percentages of CT activity [25,28,36,37,46]. Most studies applying compartmental models evaluated CT in separate compartments [22,24,27,31,32,34,35,42,46,47,48,49,51,54,55,58,59,63,65,68]. Only 12 (22.8%) studies specified the type of CT used. Forward CT was used in five studies [21,34,50,69,70]. Both backward and forward CT were evaluated in six studies [36,39,45,52,54,62].

### 3.3. Characteristics of the Models

Figure 2 illustrates the distribution of the key epidemiological parameters extracted from the included studies. Box plots show considerable variability, particularly for the infectious period, which displays the widest range across studies. In contrast, parameters such as the basic reproduction number, duration of the pre-symptomatic period, and latent period appear to be more consistent. Among the retrieved studies, 33 (62.3%) articles [20,21,22,23,24,25,26,27,28,30,31,32,35,36,39,40,42,46,47,49,52,53,54,55,56,57,58,59,61,62,63,64,67] considered quarantine and 26 (49.1%) isolation [20,21,22,23,24,25,26,27,28,30,31,32,35,36,46,47,49,52,53,54,55,56,57,62,63,64]. The basic reproduction number (R0) varies from 1.33 [30] to 4 [28,58]. The incubation period ranges from 1.5 [20] to 4 days [58]. 36 (49.1%) of the 53 studies considered the incubation period as an input parameter, and only 13 (24.5%) considered the latent period [20,22,25,30,35,38,41,44,58,64,68] as an input parameter. In 19 (35.8%) of the selected studies, the infectious period was considered an input parameter for the model.

Appendix A provides an overview of the included models. The models are categorised by approach: transmission models used are the following: 26 compartmental (49.1%) [20,22,23,24,25,27,30,31,32,34,35,42,46,47,48,49,51,54,55,58,59,60,63,64,65,68], 16 agent-based (34%) [26,28,36,37,38,40,41,43,53,56,57,61,66,67,71,72], 5 branching process model (9.4%) [21,33,44,52,62], 3 mathematical models (5.7%) [29,50,69], cluster seeding and transmission model [70], individual-level network model [64], mechanistic [73], activity driver network [45], CST [70] and Pollman et al. used both compartmental and agent-based models [39].

Compartmental models represented CT through structural changes, with new compartments introduced in most studies (e.g., Q, T, D, A, I1/I2), often to represent quarantine or detection states. Assumptions were sometimes described (e.g., partial tracing, probability-based isolation), although in many cases, they were implicit. Only a minority of these models explicitly incorporated digital CT tools, and only a few modelled forward or backward CT strategies in detail.

Agent-based models often do not introduce new compartments, relying instead on individual-level interactions and dynamic parameters to simulate CT effects. Several of these models account for real-world limitations such as imperfect compliance or delays. However, forward and backward CT are rarely explicitly modelled.

Branching process models have shown a strong focus on forward and backward tracing, with five studies explicitly modelling one or both strategies. These models often include probabilistic tracing assumptions (e.g., contact tracing with probability *p*) and examine how tracing depth or timing influences transmission. The structural complexity was generally lower, with fewer new compartments introduced.

The remaining four studies used other or hybrid modelling approaches. CT in these studies was often implemented via parameter variation (e.g., reduction of R_0_, changes in contact rates) rather than the model structure. Two of them modelled digital CT, and two reported forward or backward CT strategies.

Appendix A and Figure 3 show the different compartments considered in the articles that applied the compartmental model. The compartments reported are as follows: susceptible (S), exposed (E), infected/infective (I), recovered (R), death (D), hospitalised (H), quarantine (Q), traced (T), and other more specific compartments.

Compartment S was reported in 25 of the 26 studies considering the SEIR model, with four distinguishing susceptibility among normal, isolated, quarantined, and traced [20,24,27,58]. Compartment E was reported in 23 studies, 10 (34.2%) differentiating between normal, quarantined, traced, latent infected, and infected in untested [22,23,24,27,31,35,42,47,58,60]. The I compartment was considered in 25 (95%) studies; 17 of them differentiated infective/infected among asymptomatic, symptomatic, detected, and undetected [22,23,24,25,27,31,32,35,47,48,51,58,59,63,64,65,68]. The R compartment was reported in 24 (91.8%) studies, three of which distinguished between detected/undetected [59], tested/non-tested [55], and with immunity [22]. Compartment D was reported in 12 (45.6%) studies; three studies distinguished confirmed deaths [64], known positive or not (58), and symptomatic and asymptomatic cases [22]. Hospitalised persons were considered in seven studies [20,23,25,42,59,65,68]. Asymptomatic infection was considered in six studies [20,25,27,31,55,65], and three of them distinguished between asymptomatic super spreaders and non-super spreaders [25] and asymptomatic untested and traced [27]. Quarantine was considered in 9 studies (34.2%) [20,22,25,30,34,42,46,54,55], and traced was considered in 5 studies [20,22,27,47,48,68].

### 3.4. Infection-Related Parameters

The basic reproduction number was an input parameter in almost all the included studies. The lowest mean R_0_ considered was 1, with a range between 0.5 and 1.5 in the scenario with uniform social distancing to model the transmissibility of the virus in the German population [32]. Additionally, Hellewell et al. [33] considered a different range of R_0_ (1.5, 2.5, and 3.5), Sasmita et al. considered R_0_ between 2 and 2.5 [51], Endo assumed the value of the reproduction number between 1.2 and 2.5 [62]; Humphrey reports the tests frequencies to maintain a reproduction number of 1 [49]. Gardner and Lilpatrick examined the relationship between increased cases, delays, and reproductive number and the dynamics of SARS-CoV-2, finding that Rt follows a sigmoidal increase with increasing cases due to the loss of CT efficacy [30]. Chen et al. [25], Ashcroft et al. [21], and Pollman et al. [39] found that social distancing and CT testing can reduce R_0_ to contain epidemics.

The incubation period varies from five to six days.

The latent period in our selected studies ranged from 0.5 days [25] to 14 days [58,74]. The initial SEIR model’s latent period of 14 days was considered in the parameters [58].

The infectious period varied from 2.3 to 14 days in our study.

Quarantine was considered a specific compartment in 9 studies, distinguishing between tested or pre-symptomatic and asymptomatic. The quarantine period in almost all studies that considered it in their model was 14 days. Gardner [30] evaluated the quarantine delay as the time interval between the onset of symptoms and the start of CT, ranging from one to ten days. Kretzschmar [44] indirectly considered quarantine delay in terms of testing delay, assuming immediate isolation when testing positive; they considered a fixed four days for classical CT and 0 days for app-based CT. Kucharski [37] refers to this phenomenon as “isolation delay” and is estimated to be 2.6 days.

### 3.5. Reporting Biases

The quality assessment scores ranged from 13 to 26 of 28 (Appendix A). Nine studies (17%) were considered of very high quality, 18 studies (34%) were considered of high quality, 24 (45%) were of medium quality, and 2 (4%) were of low quality (Appendix A). The domains with the most gaps were as follows: presentation of results and uncertainty (35, 66% partial and 18, 34% yes); model structure and time horizon (15, 28% partial, 38, 72% yes); aim and objectives (8, 15% partial and 45, 85% yes), intervention/comparators (24, 45% partial, 29, 55% yes); outcome measures (19, 36% partial, 34, 64% yes); modelling methods (26, 49% partial, 27, 51% yes), assumptions explicit and justified (45, 85% partial, and 8, 15% yes) (Figure 4).

## 4. Discussion

This review provides a structured synthesis of peer-reviewed models that explicitly integrated contact tracing (CT) into the COVID-19 pandemic response. Three main themes emerged: (1) heterogeneity in model structures and CT implementation, (2) dominance of deterministic compartmental models and varying granularity, and (3) variability in infection-related parameters and their impact on outcomes.

Modelling the progression of infectious diseases has long been applied across various health domains, including influenza epidemics [75], TB vaccination impact [19], and early COVID-19 models developed up to June 2020 [76]. Among non-pharmaceutical health interventions (NPHIs), contact tracing (CT), quarantine, and screening are widely recognised as effective, particularly when implemented in combination [10].

Across the studies included, CT was modelled in different forms, with forward tracing being the most commonly adopted. Backward tracing, though less frequent, was used in studies such as the compartment model of Elias et al. [54], the agent-based study of Kerr et al. [36], the branching process of Endo et al. [62], the work by James et al. [51], and in the mathematical model of Pollman et al. [39]. This method allows the identification of superspread events, offering clear advantages in highly overdispersed epidemics, such as COVID-19 [77]. However, backward CT is generally applied to simulated settings or limited-case scenarios. For example, data from New Zealand’s EpiSurv system indicated a reduction in the effective reproduction number (Re) of up to 60% when backward tracing was applied [52]. In all these studies, CT effectiveness is valid in theory and practice but depends on the application of other interventions, such as quarantine and isolation. In all reviewed models, the effectiveness of CT was strongly dependent on contextual factors, such as timing (delay to isolation), proportion of asymptomatic transmission, and degree of transmission overdispersion.

Digital CT, although present in 30.3% of the studies included in this review, was often modelled only in theoretical terms or using agent-based simulations. dCT is an effective way to reduce the spread of SARS-CoV-2 and the costs related to manual CT. The literature on dCT is extensive and has been thoroughly reviewed in various publications, highlighting its potential benefits, challenges, ethical implications and privacy concerns. This is in line with the findings of a rapid systematic review by Jenniskens et al. [78]. Similarly, a systematic review by Pegollo et al. [79] highlighted the fragmented and heterogeneous nature of the available evidence on dCT apps, emphasising cultural, legal, and technical barriers to their adoption and the limited quantitative assessment of their effectiveness. In this review, we have omitted an in-depth discussion of dCT due to the distinctive nature of this topic, which warrants dedicated attention beyond the scope of our current review.

Several studies have also evaluated the effectiveness of CT using various assumptions. Colomer et al. [28] ranged the efficacy of CT from 0 to 40% to evaluate its importance in two scenarios, with and without social distancing measures. Their findings endorse the efficacy of CT as a valuable tool in managing the pandemic trajectory, aligning with similar conclusions from other published studies, even when CT was performed at a modest (40%) rate. They also highlight a major drawback of CT, emphasising the substantial testing requirements and the associated high costs for the government. Gardner et al. [30] considered the efficacy of the fraction of infected individuals traced. Kucharski et al. [37] assumed that the proportion of potentially traceable contacts that are successfully traced is 100% in a home setting and 95% in a workplace or school setting.

### 4.1. Stochastic and Deterministic

Mathematical models applied to the epidemiology of infectious diseases, such as SARS-CoV-2, can be classified into deterministic or stochastic models. Deterministic models consider non-random rate flows in a population stratified into compartments, while stochastic models assume random variations in the movements between compartments [80]. The predominant modelling approach in the included studies was the deterministic model based on the susceptible-exposed-infectious-recovered (SEIR) framework for epidemic modelling. This aligns with the findings of Shankar et al. [76], who conducted a systematic review of the initial mathematical models used to explain the spread of COVID-19. Deterministic models for infectious diseases aim to approximate epidemic dynamics within a closed system. These models compartmentalise the population by employing differential equations to describe transitions between states (compartments) and capture variations over time [80].

Various models considering the SEIR model have attempted to adapt it to different epidemiological situations by adding new compartments. Considering that we included models that considered CT, the compartments were those of quarantined people, traced, hospitalised, dead, and asymptomatic/infected. Questions related to confirmed diagnoses are considered in the infected/infectious compartment. In some compartments, the authors distinguished between traced, tested, untested, and asymptomatic. Especially with the spread of COVID-19 cases, the spread of the virus through asymptomatic people and consequently undetected cases has been of great concern. However, although deterministic models capture the epidemiological behaviour of infectious diseases, they do not consider the frequent random events that can influence transmission dynamics. Deterministic models are used mainly when the population size is large, and as shown in our review, they were applied in populations greater than 32,583 [42] in China. Only Hu et al. [64] used a compartmental model in a sample size population of 3000 people. As suggested in the literature, stochastic models, such as branching processes and agent-based models, have been used in regions with small populations.

Agent-based models (ABM), also known as individual-based models (IBM), are bottom-up methodologies [81]. Each individual is considered an agent with specific characteristics. ABMs can produce emergent macroeffects from microrules [82]. Epidemiological applications of ABM are designed to perform a preliminary analysis to assess system behaviour under various conditions and to assess which policies to adopt to combat epidemics [83]. The selected studies mainly applied agent-based and branching process models to simulate populations. Only James et al. [52] used a branching process model with 101 participants.

### 4.2. Infection-Related Parameters

The effectiveness of COVID-19 transmission models is influenced by the assumptions made regarding key infection-related parameters. Among these, the basic reproduction number (R_0_), incubation period, latent period, infectious period, and quarantine duration were considered. The Basic Reproduction Number (R_0_) is an epidemiological measure used to quantify the contagiousness of individuals. It represents the average number of secondary cases generated by an infected person in a susceptible population [84]. It is a cornerstone parameter for assessing the potential spread of an infectious disease and designing appropriate containment strategies. The reviewed studies reported considerable variability in the R_0_ values, reflecting differences in the model assumptions and settings. While this review does not specifically account for the influence of SARS-CoV-2 variants on R_0_, the existing literature suggests that viral evolution may contribute significantly to the observed heterogeneity [85].

The incubation period is defined as the time between exposure and symptom onset. It is a crucial epidemiological measure to curtail the spread of infectious diseases in the surveillance, monitoring, and modelling of infectious diseases [86]. Knowledge of the incubation period helps define quarantine policies and other interventions [87]. The incubation period in the studies ranged from five to six days, which is consistent with the findings in the literature. Dhouib et al. [88] reported a mean incubation period between 5.2 (95% CI 4.4, 5.9) and 6.65 days (95% CI 6.0, 7.2). In the systematic review and meta-analysis by Rai et al., the pooled incubation period was 5.74 (5.18, 6.30) [86]. A global meta-analysis of 53 studies by Cheng et al. found that the COVID-19 incubation period was 6.0 days (95% CI 5.6, 6.5) [89]. recent studies have highlighted the variability introduced by different SARS-CoV-2 strains. For instance, a systematic review and meta-analysis by Wu et al. showed an average incubation period of 6.57 days, which became shorter with the emerging Delta variants (pooled incubation period of 4.41 days, 95% CI 3.76, 5.05 days) and Omicron (pooled incubation period of 3.42 days, 95% CI 2.88, 39 days) [90].

The latent period is the time between exposure and the onset of infectiousness. Unlike the incubation period, which refers to the time before symptom onset, the latent period focuses on infectiousness, which can begin before symptoms appear or even in asymptomatic cases. In the models included in this review, the latent period ranged from 0.5 days [25] to 14 days [58,74]. The upper limit of 14 days was used in SEIR models as a precautionary assumption before detailed virological data became available [58]. More recent and data-driven estimates clustered around a mean of 5.5 days (95% CI: 5.1–5.9) as reported by Zhang et al. [74]. The infectious period represents the time between the beginning of the infection and the non-infectious period. In our study, the range was from 2.3 days to 14 days. This variation may reflect the differing assumptions about viral load dynamics and case management. Shorter infectious periods are typically assumed for milder or asymptomatic cases, whereas longer durations may apply to unisolated or more severe cases. Differences between variants may also play a role, as viral load kinetics are known to differ between strains [91].

Quarantine duration was frequently modelled as a fixed period—most often 14 days—based on early public health guidance. Some models incorporate delays between symptom onset and quarantine initiation, capturing more realistic implementation scenarios. Although a 14-day quarantine period has become the standard in many countries, its statistical justification remains complex. A likelihood-based analysis by Charvadeh et al. [92] questioned the sufficiency of this duration, suggesting that 14 days may not always be long enough to minimise the risk of premature release of infected individuals.

### 4.3. Study Limitations

Although we provide a comprehensive review of the models reported in the literature that use CT, we cannot compare the estimates with each other because they were applied to different populations. Future development of this work could involve implementing all the models included in this review in a unique population and comparing their different performances. This could help define the information needed to accurately predict the COVID-19 pandemic.

## 5. Conclusions

Our research adds value to the existing literature by providing a comprehensive review of models that specifically incorporate contact tracing in the context of the COVID-19 pandemic and its variants. Although other publications and reports have explored related topics, our study stands out for its focus on the role and effectiveness of contact tracing within mathematical models. This modelling perspective is particularly relevant given the evolving dynamics of the pandemic and the need for adaptable public health strategies.

CT is an effective public health tool for infectious disease control. The review by Afzal et al. [13] reported that it is an effective tool for various diseases, such as COVID-19. However, there is significant heterogeneity in the approaches used in different countries [6]. CT strategies should be adapted considering the epidemic scenario, viral evolution, and transmission patterns to ensure efficient and sustainable management.

Mathematical modelling has played a key role in understanding the epidemiological trajectory of COVID-19 [6]. The inclusion of the NPHI in these models is helpful in the various phases of the disease, as it allows for the evaluation of the effects of different measurements on epidemiological development. Including CT in the model for modelling or predictive purposes helps evaluate the effectiveness of interventions. However, assessing the magnitude of these effects requires another type of consideration to define CT goals in different epidemiological situations and the most cost-effective way to implement them.

This study systematically synthesises and analyzes a diverse set of mathematical models, offering a comprehensive overview of the various approaches used in literature. We provide insights into the effectiveness of CT under different assumptions, including population size, intervention measures, and integration of digital contact tracing.

By categorising and summarising the models based on their structures, infection-related parameters, and reporting biases, we offer a valuable resource for researchers and policymakers seeking a deeper understanding of the modelling landscape. Considering only mathematical and statistical models has caused some difficulties in evaluating the impact of single models. There is great heterogeneity in the types of models, and each starts from different assumptions on relevant model parameters. This systematic qualitative review provides a comprehensive view of the models proposed to evaluate the COVID-19 pandemic, including the NPHI.

## Figures and Tables

**Figure 1 healthcare-13-00935-f001:**
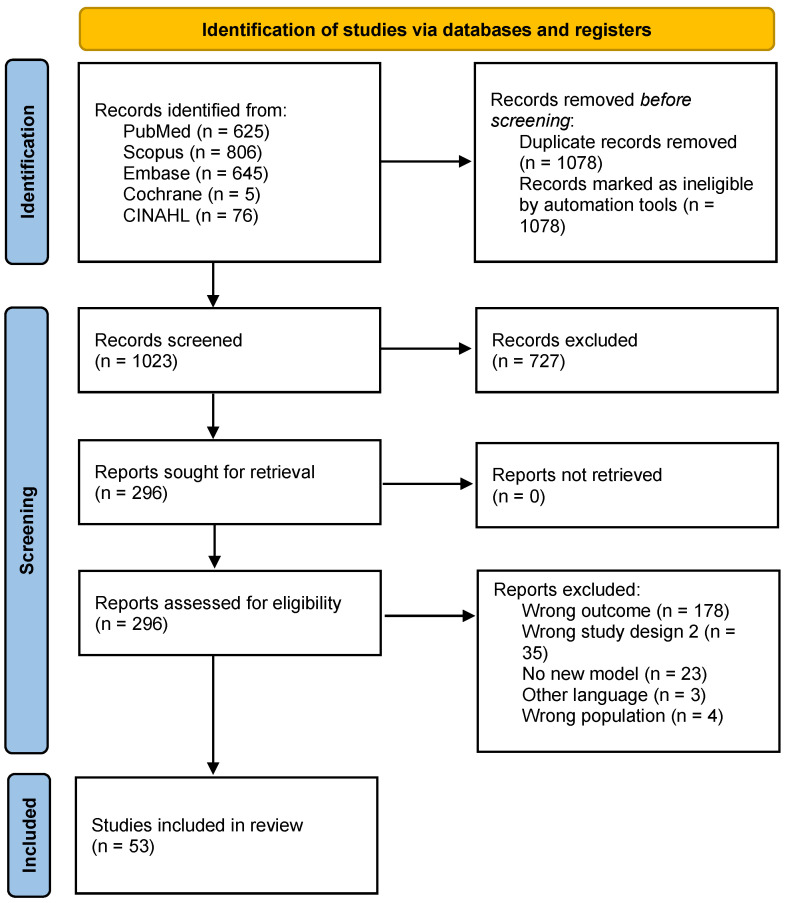
PRISMA 2020 flow diagram.

**Figure 2 healthcare-13-00935-f002:**
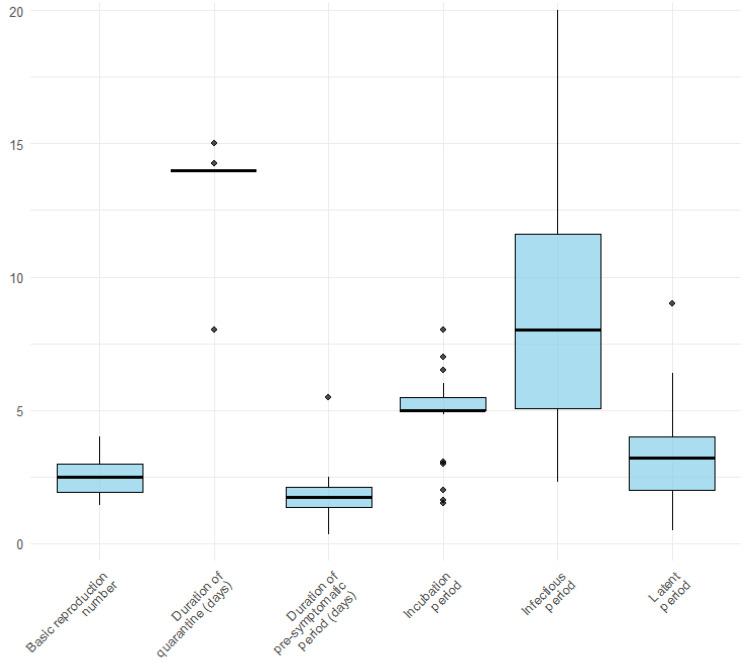
Variation in model parameters used in the included studies.

**Figure 3 healthcare-13-00935-f003:**
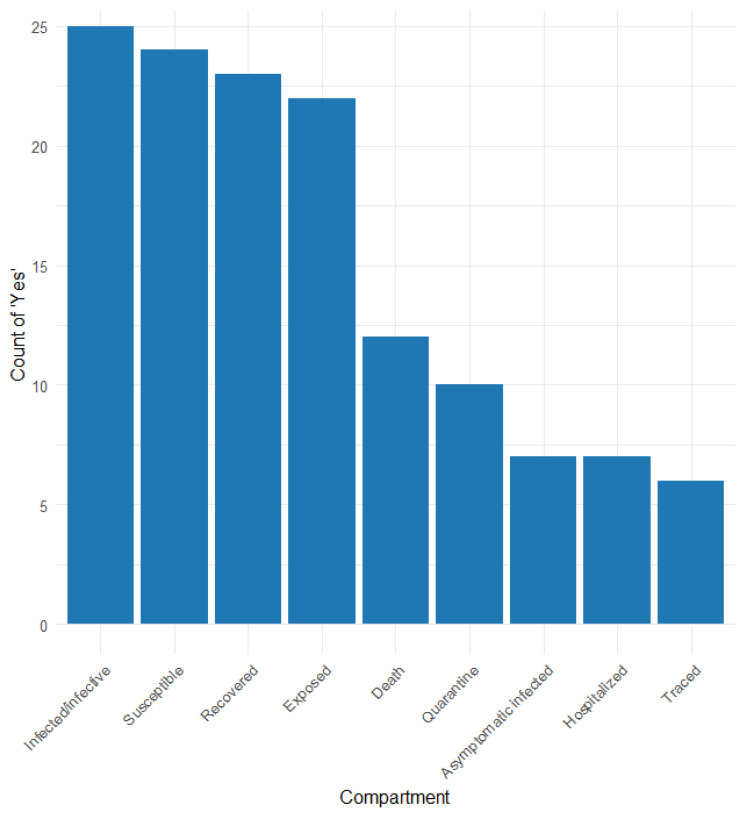
Frequency of compartment types used in contact tracing models.

**Figure 4 healthcare-13-00935-f004:**
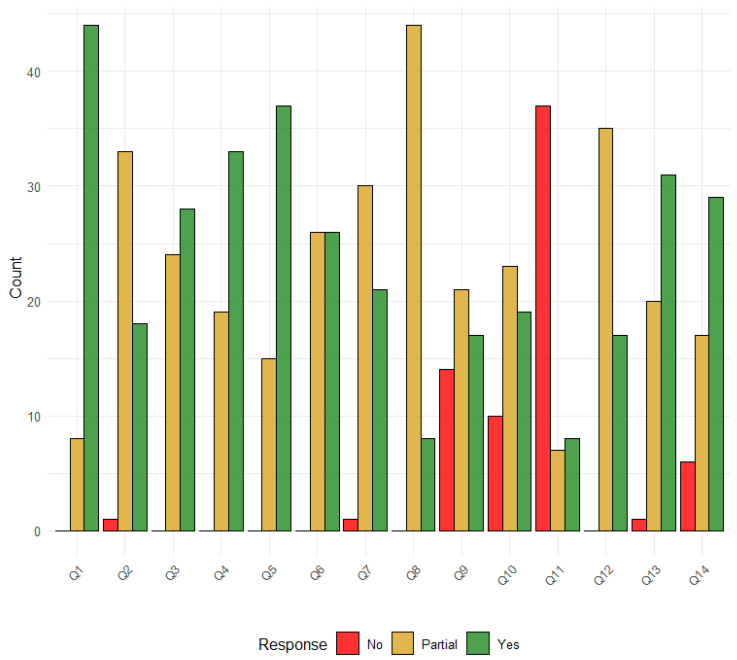
Distribution of Risk of Bias Responses. Legend: Q1: Aim and objectives; Q2: Setting and population; Q3: Intervention comparators; Q4: Outcome measures; Q5: Model structure and time horizon; Q6: Modelling methods; Q7: Parameter ranges and data sources; Q8: Assumptions explicit and justified; Q9: Quality of data and uncertainty or sensitivity analysis; Q10: Method of fitting; Q11: Model validation; Q12: Presentation of results and uncertainty; Q13: Interpretation and discussion of results; Q14: Funding source and conflicts of interest.

## Data Availability

The data supporting the findings of this study are available upon reasonable request from the corresponding author.

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
