# Peer review of "Mathematical Contact Tracing Models for the COVID-19 Pandemic: A Systematic Review of the Literature"

_healthcare, 2025, doi:10.3390/healthcare13080935_

Round 1

Reviewer 1 Report

Comments and Suggestions for Authors

The paper presents a literature review of mathematical and computational models that studied the impact of contact tracing interventions during the Coronavirus pandemic. The paper is well written and the work can contribute to the body of knowledge in public health modelling.   

I have a couple of comments with regard the methods:

-What are the years of publication that are included in the search?

-What are the keywords?

-From the dates, I understand that the work is quite dated. I would like to see a justification for this belated publication.

Also, the are some minor presentation quality issues:

-page 1, line 38 – wrong date -> “On 23 December 2022, the novel Coronavirus, first identified in Wuhan …” it is 2020.

-In text citation using both parenthesis () and angle bracket [] -> please use one or the other for consistency.

-References – news items accessed back in 2022 (e.g. [4] … (accessed on 24 May 2022).) – please update.

Author Response

We thank the reviewer for their valuable comments, which helped us improve the clarity and depth of the manuscript. We have carefully revised the text to better highlight the key findings and provide a more structured and informative discussion. 

( ) I would not like to sign my review report 
(x) I would like to sign my review report 

Quality of English Language 

( ) The English could be improved to more clearly express the research. 
(x) The English is fine and does not require any improvement. 

Yes 

Can be improved 

Must be improved 

Not applicable 

Does the introduction provide sufficient background and include all relevant references? 

(x) 

( ) 

( ) 

( ) 

Is the research design appropriate? 

( ) 

( ) 

( ) 

( ) 

Are the methods adequately described? 

( ) 

(x) 

( ) 

( ) 

Are the results clearly presented? 

(x) 

( ) 

( ) 

( ) 

Are the conclusions supported by the results? 

(x) 

( ) 

( ) 

( ) 

Comments and Suggestions for Authors 

The paper presents a literature review of mathematical and computational models that studied the impact of contact tracing interventions during the Coronavirus pandemic. The paper is well written and the work can contribute to the body of knowledge in public health modelling.    

I have a couple of comments with regard the methods: 

-What are the years of publication that are included in the search? 

Thank you for your comment. 

The search included studies published up to May 15th, 2022, which reflects the end of our data collection phase. This data was kept as it approximately coincides with the end of the acute phase of the COVID-19 pandemic in many countries, when large-scale interventions such as contact tracing began to be phased out or significantly modified.  

 -What are the keywords? 

Thank you for your comment. As mentioned in the manuscript, the search strategy was based on three core concepts: COVID-19, contact tracing, and statistical/mathematical models. Keywords within each concept were combined using the OR Boolean operator, and the three groups were connected using AND. The full list of keywords and their combinations is provided in in Appendix A, Table A1. We have clarified within the text.  

-From the dates, I understand that the work is quite dated. I would like to see a justification for this belated publication. 

Thank you for raising this point. While the search was completed in May 2022, the subsequent phases of study selection, data extraction, and critical synthesis required substantial time due to the volume and heterogeneity of the included studies. We also aimed to ensure methodological rigor and clarity in reporting. Importantly, the search period covers the core phase of the COVID-19 pandemic, during which most contact tracing strategies were implemented and most modelling studies were published. For this reason, we believe the review still provides valuable and timely insights for both retrospective evaluation and future preparedness planning. 

Also, the are some minor presentation quality issues: 

-page 1, line 38 – wrong date -> “On 23 December 2022, the novel Coronavirus, first identified in Wuhan …” it is 2020. 

Thanks, revised. 

-In text citation using both parenthesis () and angle bracket [] -> please use one or the other for consistency. 

Thanks, revised. 

-References – news items accessed back in 2022 (e.g. [4] … (accessed on 24 May 2022).) – please update. 

Thanks, revised. 

Reviewer 2 Report

Comments and Suggestions for Authors

This work is a systematic review of the literature on mathematical contact tracing models for the COVID-19 pandemic. 

I have read the entire manuscript thoroughly. Although it is well written and the methodology and results seem correct, there are two minor points that need to be included in the work before publication.

1- Firstly, the authors must include the key words of the research at the beginning of the material and methods. 
2- Why did the authors limit their search to the databases listed? 

Author Response

We thank the reviewer for their valuable comments, which helped us improve the clarity and depth of the manuscript. We have carefully revised the text to better highlight the key findings and provide a more structured and informative discussion. 

Open Review 

(x) I would not like to sign my review report 
( ) I would like to sign my review report 

Quality of English Language 

( ) The English could be improved to more clearly express the research. 
(x) The English is fine and does not require any improvement. 

Yes 

Can be improved 

Must be improved 

Not applicable 

Does the introduction provide sufficient background and include all relevant references? 

( ) 

(x) 

( ) 

( ) 

Is the research design appropriate? 

( ) 

(x) 

( ) 

( ) 

Are the methods adequately described? 

(x) 

( ) 

( ) 

( ) 

Are the results clearly presented? 

(x) 

( ) 

( ) 

( ) 

Are the conclusions supported by the results? 

(x) 

( ) 

( ) 

( ) 

Comments and Suggestions for Authors 

This work is a systematic review of the literature on mathematical contact tracing models for the COVID-19 pandemic.  

I have read the entire manuscript thoroughly. Although it is well written and the methodology and results seem correct, there are two minor points that need to be included in the work before publication. 

1- Firstly, the authors must include the key words of the research at the beginning of the material and methods.  

Thanks, we have revised the paragraph related to the search strategy.  
2- Why did the authors limit their search to the databases listed?  

Thank you for your observation. We acknowledge that additional databases such as IEEE Xplore, ACM Digital Library, or arXiv may index modelling studies developed primarily within the fields of computer science or theoretical mathematics. However, our primary objective was to capture studies that applied modelling approaches to evaluate contact tracing interventions in the context of the COVID-19 pandemic, with a strong emphasis on public health and biomedical relevance. For this reason, we focused on major health and multidisciplinary databases—such as PubMed, EMBASE, CINAHL, Cochrane Library, and Scopus—which together provide a wide coverage of both applied modelling studies and epidemiological literature. 

Submission Date 

11 March 2025 

Date of this review 

21 Mar 2025 11:33:46 

Reviewer 3 Report

Comments and Suggestions for Authors

My comments are as follows:
1. Tables 1–4 are highly unfriendly to readers. On one hand, their readability is extremely poor; on the other hand, presenting raw data without any processing reflects inadequate data handling skills and a lack of scientific rigor.  
2. In Section 3 ("Results"), nearly all the text consists of simple counting or basic statistical reporting. This approach is not appropriate for a serious scientific manuscript.  
3. The language used in this paper is often unnatural and, at times, confusing. For example, the sentence “Finally, Kucharski (36) called this phenomenon the isolation delay and the first considered a lap of 2.6 days.” (lines 355–356) is unclear and grammatically incorrect.  
4. The terms "Forward CT" and "Backward CT" are used frequently without any prior definition, which reflects a lack of fundamental scientific rigor.  
5. The "Discussion" section does not provide any meaningful insights or conclusions that would be valuable to readers.  
6. Given that this is a review on contact tracing, it is essential to compare it with other relevant reviews on the topic.  

Comments on the Quality of English Language

The language used in this paper is often unnatural and, at times, confusing. For example, the sentence “Finally, Kucharski (36) called this phenomenon the isolation delay and the first considered a lap of 2.6 days.” (lines 355–356) is unclear and grammatically incorrect.  

Author Response

We thank the reviewer for their valuable comments, which helped us improve the clarity and depth of the manuscript. We have carefully revised the text to better highlight the key findings and provide a more structured and informative discussion. 

Open Review 

(x) I would not like to sign my review report 
( ) I would like to sign my review report 

Quality of English Language 

(x) The English could be improved to more clearly express the research. 
( ) The English is fine and does not require any improvement. 

Yes 

Can be improved 

Must be improved 

Not applicable 

Does the introduction provide sufficient background and include all relevant references? 

( ) 

( ) 

(x) 

( ) 

Is the research design appropriate? 

( ) 

( ) 

( ) 

(x) 

Are the methods adequately described? 

( ) 

( ) 

( ) 

(x) 

Are the results clearly presented? 

( ) 

( ) 

(x) 

( ) 

Are the conclusions supported by the results? 

( ) 

( ) 

( ) 

(x) 

Comments and Suggestions for Authors 

My comments are as follows: 
1. Tables 1–4 are highly unfriendly to readers. On one hand, their readability is extremely poor; on the other hand, presenting raw data without any processing reflects inadequate data handling skills and a lack of scientific rigor.   

We thank the reviewer for the feedback. We acknowledge the importance of presenting information in a clear and reader-friendly way. In response, we have revised the layout of Tables 1–4 to improve readability and consistency. However, we would like to clarify that reporting the raw model characteristics was a deliberate choice in line with PRISMA recommendations and standard practice in systematic reviews, especially when summarizing heterogeneous modelling frameworks. These tables aim to provide transparency and allow readers to evaluate the individual features of each included study. 
Moreover, the manuscript already includes several forms of data processing and synthesis. In the Results section, we summarize proportions of model types, epidemiological parameter ranges (e.g., R₀, incubation period), and the frequency of key structural assumptions (e.g., compartmental additions, use of real vs. simulated data). These elements were derived from the detailed tables but presented in an aggregated and comparative manner to support interpretation. 
2. In Section 3 ("Results"), nearly all the text consists of simple counting or basic statistical reporting. This approach is not appropriate for a serious scientific manuscript.   

We thank the reviewer for this important observation. 
We respectfully clarify that, in accordance with the standards of systematic reviews (e.g., PRISMA guidelines), the Results section is intended to provide a structured and descriptive synthesis of the included studies, without interpretation or evaluation, which are reserved for the Discussion section. 

To improve clarity and readability, we have: 

Moved the tables in the Supplementary Materials (Tables S1–S3); 

Added three new figures (Figures 2–4) to visually summarize the distribution of study types, compartments, and model assumptions. 

  1. The language used in this paper is often unnatural and, at times, confusing. For example, the sentence “Finally, Kucharski (36) called this phenomenon the isolation delay and the first considered a lap of 2.6 days.” (lines 355–356) is unclear and grammatically incorrect.  

We thank the reviewer for highlighting issues with the clarity and fluency of the language. 
In response, we have thoroughly revised the manuscript for language and style. We used Grammarly to assist with grammar and clarity improvements, and we also performed a detailed manual review of all sections to ensure the text reads fluently and professionally. 

  1. The terms "Forward CT" and "Backward CT" are used frequently without any prior definition, which reflects a lack of fundamental scientific rigor.  

Thanks for pointing this out, we have added a paragraph of explanation in the introduction. 

  1. The "Discussion" section does not provide any meaningful insights or conclusions that would be valuable to readers.  

We thank the reviewer for this observation. In response, we have substantially restructured and expanded the Discussion section to provide a clearer synthesis of the key findings and their implications. 

  1. Given that this is a review on contact tracing, it is essential to compare it with other relevant reviews on the topic.  
    We have added a concise comparison with other relevant systematic reviews, notably those by Jenniskens et al. and Pegollo et al., to contextualize our findings and clarify the contribution of our review. 

Comments on the Quality of English Language 

The language used in this paper is often unnatural and, at times, confusing. For example, the sentence “Finally, Kucharski (36) called this phenomenon the isolation delay and the first considered a lap of 2.6 days.” (lines 355–356) is unclear and grammatically incorrect.   

Submission Date 

11 March 2025 

Date of this review 

26 Mar 2025 15:10:00 

Fine modulo 

© 1996-2025 MDPI (Basel, Switzerland) unless otherwise stated 

Reviewer 4 Report

Comments and Suggestions for Authors

The manuscript compiles mathematical models of COVID-19 that include contact tracing. While the authors have done a good job compiling the models and statistics about the models, presentation of the work is rather superficial. Specific comments:

  1. While the authors present all kinds of basic facts about the model, there is really no synthesis or overall takeaway from the manuscript. I would appreciate a more in depth exploration of how model predictions compare and why they might differ.
  2. The authors should explain the different contact tracing methods mentioned in the manuscript (forward, backward, digital, etc.)
  3. Presenting some of the data in figures or graphs would be helpful --- the tables are large and contain way too much information to parse easily. For example, perhaps R0, infectious period, etc. could be presented as distributions, boxplots, or violin plots.
  4. What are "wrong outcomes"?

Author Response

We thank the reviewer for their valuable comments, which helped us improve the clarity and depth of the manuscript. We have carefully revised the text to better highlight the key findings and provide a more structured and informative discussion. 
Open Review 

(x) I would not like to sign my review report 
( ) I would like to sign my review report 

Quality of English Language 

( ) The English could be improved to more clearly express the research. 
(x) The English is fine and does not require any improvement. 

Yes 

Can be improved 

Must be improved 

Not applicable 

Does the introduction provide sufficient background and include all relevant references? 

(x) 

( ) 

( ) 

( ) 

Is the research design appropriate? 

(x) 

( ) 

( ) 

( ) 

Are the methods adequately described? 

(x) 

( ) 

( ) 

( ) 

Are the results clearly presented? 

( ) 

(x) 

( ) 

( ) 

Are the conclusions supported by the results? 

( ) 

(x) 

( ) 

( ) 

Comments and Suggestions for Authors 

The manuscript compiles mathematical models of COVID-19 that include contact tracing. While the authors have done a good job compiling the models and statistics about the models, presentation of the work is rather superficial. Specific comments: 

  1. While the authors present all kinds of basic facts about the model, there is really no synthesis or overall takeaway from the manuscript. I would appreciate a more in depth exploration of how model predictions compare and why they might differ. 

We appreciate this comment and have revised the Discussion to include a more structured synthesis of the findings, organized around three cross-cutting themes: model structure, infection-related parameters, and variability in CT implementation. We also now explicitly discuss how differences in assumptions (e.g., R₀, latency, overdispersion, digital/manual CT use) can lead to divergent model predictions, particularly in terms of the estimated effectiveness of CT. 

2.The authors should explain the different contact tracing methods mentioned in the manuscript (forward, backward, digital, etc.) 
Thanks for pointing this out, we have added a paragraph of explanation in the introduction. 

3.Presenting some of the data in figures or graphs would be helpful --- the tables are large and contain way too much information to parse easily. For example, perhaps R0, infectious period, etc. could be presented as distributions, boxplots, or violin plots. 

We thank the reviewer for this important observation. 
To improve clarity and readability, we have: 

Moved the tables in the Supplementary Materials (Tables S1–S3); 

Added three new figures (Figures 2–4) to visually summarize the distribution of study types, compartments, and model assumptions. 

4.What are "wrong outcomes"? 
Thanks for pointing this out, we have clarified in the methods section. 

Submission Date 

11 March 2025 

Date of this review 

30 Mar 2025 06:22:48 

Fine modulo 

© 1996-2025 MDPI (Basel, Switzerland) unless otherwise stated 

Round 2

Reviewer 3 Report

Comments and Suggestions for Authors

This revised manuscript has exceeded my expectations in terms of improvement. Although the formatting of the paper currently appears somewhat unprofessional, I believe this issue can be addressed by a professional editing team. Given this situation, I think the likelihood of further enhancing the quality of the paper is relatively low. Therefore, I recommend the publication of this manuscript.

Author Response

We sincerely thank the reviewer for the positive and encouraging feedback on our revised manuscript. We are particularly pleased to hear that the improvements have exceeded expectations.

 We are committed to addressing the issue related to the the formatting during the final production phase to ensure the paper meets the highest publication standards.

Thank you again for your thoughtful review and recommendation for publication.

Reviewer 4 Report

Comments and Suggestions for Authors

The manuscript is much improved. I would only suggest that the authors careful review the added sections as I noted a few typos.

Author Response

We sincerely thank the reviewer for the positive feedback.

We appreciate the suggestion regarding minor typos in the newly added sections. Following your advice, we have carefully reviewed the entire manuscript and corrected the typographical errors to ensure clarity and consistency throughout the text.